# Interregional circulation of credit resources and company innovation

Xinglin Liu[1]*, Yu Chen[2], Yan Xiong[3], Yanlin Wu[4]

**1** School of Economics, Shenzhen Polytechnic University, Shenzhen, China, **2** School of Management, Wuhan Institute of Technology, Wuhan, China, **3** School of Science, Hubei University of Technology, Wuhan, China, **4** School of Humanities, Guangdong Peizheng College, Guangzhou, China

* liuxinglin2018@126.com

**Data Availability Statement:** All relevant data are within the manuscript and its Supporting information files.

**Funding:** This work was supported by the Shenzhen Polytechnic University Research Fund (6024310010S to X.L.).

## Abstract

Although the impact of interest rates, repayment periods, and loan scales on loan consequences has been extensively studied, little attention has been paid to the geographical distance involved in loan transactions. This study collects the addresses of borrowing companies, listed companies, and banks. Nonlocal loans can be distinguished because the regional segmentations in the lending industry reflect the features of provincial boundaries. Using data from Chinese A-share listed companies from 2007 to 2022, this research explores the causes of nonlocal loans and their impact on company innovation. Nonlocal loans are found to address the lack of local credit resources rather than financial constraints, supplementing disposable capital. This interregional circulation of credit resources facilitates innovation, particularly in financially undeveloped areas. This study does not detect research and development manipulation and recognizes the increase in innovation output. The findings have implications for credit resource allocation and balanced regional development.

## Introduction

Market segmentation is a significant barrier to resource allocation and regional development [1, 2], obstructing market integration [3]. Hammer [4] advocates the elimination of market segmentation to facilitate economic growth and promote interregional cooperation. There is also evidence that interregional flows alleviate local economic fluctuations [5]. In February 2020, China issued the Guideline on Accelerating the Construction of Shanghai International Financial Center and Providing Financial Support for the Integrated Development of the Yangtze River Delta (hereafter referred to as the Guideline), which encourages the cross-provincial allocation of credit resources and aims to balance and meet the credit needs of companies in various provinces. Using this background as the foundation, this paper explores the interregional circulation of credit resources in China.

Past studies have highlighted the key role of geographical, economic, and institutional factors in shaping the direction and volume of interregional credit flows [3, 6]. Regions with greater economic activity, more developed financial markets, and stronger banking sectors tend to be net exporters of credit, whereas less developed regions are typically net importers

**Competing interests:** The authors have declared that no competing interests exist.

[7]. Informational asymmetries and transaction costs associated with lending across regions can also restrict the flow of credit. Advancements in credit-scoring models have helped to decrease loan default risks [8], enabling banks to cooperate with companies located at greater distances [9]. Furthermore, the rapid progress in information technology in recent decades has stimulated the interregional circulation of credit resources [10]. However, geographic distance remains a relevant factor in loan transactions [11]. While it has become significantly easier to collect hard information [12], the transmission of soft information generally requires face-to-face interaction between borrowers and lenders [13]. The uncertainty about soft information caused by a lack of interaction can result in lower loan amounts and higher interest rates for interregional credit flows [14, 15].

While existing research efforts have provided detailed explanations of the effect of geographic distance on loan terms, there are still two significant research gaps. Firstly, the economic impact of loans over long distances has not been identified. While the effect of loans on innovation in terms of interest rate, loan scale, repayment period, and credit market accessibility has been explored extensively [16, 17], geographic distance has received limited consideration. Information costs force banks to set higher interest rates for loans when the borrowers are further away, which may increase the operating burden of companies and discourage innovation [18]. Companies are less likely to establish remote partnerships if loan transactions are expected to reduce financial performance or inhibit innovation. As research and development (R&D) projects require substantial and stable financial support, supplementary loans are likely required to meet the financing needs of companies in financially undeveloped areas [19]. Therefore, consideration of the impact of cross-regional loans is valuable. Secondly, although the shortage of remote loan businesses has been emphasized, the reasons why there is substantial demand for loans have not been discussed in depth. Huang et al. [20] argue that the long-distance loan industry has been created by the development of digital finance, and some studies suggest that cross-regional lending activities usually involve companies with high social trust [21, 22]. However, the occurrence of interregional circulation of credit resources has not been discussed from the demand perspective. Further studies could consider the influence of the economic environment, financing restraints, or innovation demand.

To fill these gaps, this paper explores why companies pursue interregional credit resources and the effect of these loans on company innovation. In terms of regional segmentation, the loan industry is reflective of provincial boundaries rather than city boundaries in China [23–25]. Consistent with the Guideline, credit resources from provinces in which the company, its parent, or its subsidiary is not located are regarded as nonlocal loans in this paper. Using a sample of Chinese A-share listed companies from 2007 to 2022, this study demonstrates that companies with a transfusion of nonlocal loans are willing to increase their R&D investment intensity. The discussion part of this paper points out that the main reason why companies seek nonlocal loans is the shortage of local credit resources. Rather than alleviating repayment pressures or reducing financial constraints, nonlocal loans fill the geographic gap in the stock of credit resources and stimulate enthusiasm for innovation among companies in financially undeveloped areas.

The contributions of this paper are as follows. Firstly, it enriches the research on the economic consequences of loans with regard to company innovation. It has been proven that the effect of loans on company innovation is influenced by the interest rate, loan scale, and repayment period [16, 26], yet geographic factors are seldom mentioned. This paper broadens the research scope by dividing loans into local and nonlocal ones with provincial boundaries. The characteristics of loans and their effect are also explored. Secondly, this paper clarifies the function of nonlocal loans as supplementary capital. While the existing literature highlights the negative influence of long distances between companies and banks on loan agreements [27, 28], the

positive effect of nonlocal loans has been ignored. Referring to resource dependence theory, this study provides a new perspective on financing channels through an exploration of how companies cope with insufficient local credit resources. The findings explain why companies are willing to take on nonlocal loans despite the higher interest rates and longer distances involved. Moreover, this study rejects the application of the polarization effect theory in the Chinese lending industry by demonstrating that the interregional circulation of credit resources benefits companies in financially undeveloped areas instead of financially developed areas. The results are helpful in objectively understanding the role of nonlocal loans. Since some findings present the spatial agglomeration and Matthew effect for innovation output [29, 30], this paper puts forward a possible solution for the polarization dilemma. Credit resources should be allocated appropriately to enhance high-quality economic development in undeveloped areas.

## Hypothesis development

### Financial development level and nonlocal loans

The resource dependence theory posits that organizations are interdependent with their external environment [31]. They can adjust the degree of this dependence through various strategies. In the context of corporate financing, companies may seek nonlocal bank loans if local banking institutions are unable to provide sufficient credit resources for their operational or innovative requirements. Existing research supports this notion; for example, Lai et al. [32] show that there is a close relationship between companies' R&D investment intensity and the financial development level of the province in which they are located. The more accessible the credit resources, the easier it is for companies to finance their innovation activities. Furthermore, Shi et al. [33] demonstrate that there is a higher threshold for economic development in the western region compared to the eastern region of China. Moreover, according to the Report on China Regional Finance in 2021, the deposit and loan balances of financial institutions have a highly uneven geographical distribution, with the eastern, central, western, and northeastern regions accounting for 55.8%, 17.8%, 20.5%, and 5.9%, respectively. This evidence suggests that companies in financially underdeveloped provinces are at a disadvantage in terms of accessing the credit needed to fund innovation. However, Cumming et al. [34] argue that the financial gap between regions can be narrowed through nonlocal loans. Consequently, the following hypothesis is proposed:

**Hypothesis 1**. Companies in financially undeveloped provinces are more likely to pursue nonlocal loans.

### Financing constraints and nonlocal loans

When confronted with financial constraints, companies may attempt to borrow from nonlocal banks to broaden their financing channels. This concept is supported by existing research; for example, Degryse et al. [27] show that companies that suffer from severe financial constraints can benefit from intense bank competition. This competition results in higher risk tolerance among banking institutions, thus increasing the overall availability of credit resources for companies [35, 36]. To relieve their capital shortage, companies may be motivated to seek agreements with nonlocal banks located in provinces with a more competitive banking environment. However, Huang et al. [37] argue that financial constraints can make it difficult for companies to obtain financial support from both local and nonlocal banks. Furthermore, the cross-provincial lending industry may not be a viable option for the most financially constrained firms [38]. Given these perspectives, the second hypothesis is as follows:

**Hypothesis 2**. Companies with financial constraints are more likely to pursue nonlocal loans.

## Nonlocal loans and R&D investment intensity

Diamond [39] highlights the role of banks as regulators and confirms their influence on corporate decision-making. However, geographic distance can limit the ability of banks to oversee their operations efficiently [40]. From a risk compensation perspective, the higher regulatory costs associated with long-distance loan transactions reflect the greater repayment risk [9]. Remote locations can also render bank supervision and restrictions less binding, allowing borrowers more leeway to pursue risky investments such as R&D projects [8].

Existing research suggests that interregional loan activity can help compensate for local credit shortages and promote the development of regional economies [41]. According to Stratopoulou [42], companies in economically underdeveloped areas can benefit from government grants and nonlocal loans, with the positive effect of the latter being particularly significant. To mitigate the information asymmetry associated with long distances and remote loan transactions, banks tend to favor lending to companies with high analyst ratings and transparent financial reporting [43]. For these well-performing borrowers, access to nonlocal credit resources can provide a crucial boost for their innovation activities. Based on the above analysis, the following hypothesis is given:

**Hypothesis 3a**. Nonlocal loans have a positive effect on R&D investment.

Banks generate interest income from their lending activities but do not directly profit from the innovative achievements of borrowing companies [44]. These institutions must also bear the repayment risk, especially when R&D projects fail. This mismatch between risk and return may prompt banks to closely monitor and intervene in the capital usage of their borrowers [38]. Technological advancements have made it more convenient for banks to collect company-related data, extending their supervisory reach [40]. As explained by Malkonen and Vesala [28], when providing credit services, banks implement differential pricing strategies based on the costs of information, transactions, and supervision. Furthermore, Hollander and Verriest [45] argue that banks will gain a dominant position in loan agreements if geographic factors exacerbate information asymmetries. A greater geographic distance between borrowers and lenders leads to higher interest rates [46], which in turn may force borrowers to prioritize daily operations over R&D projects.

It is important to note that borrowing from nonlocal banks instead of local banks could be indicative of financial constraints. Because banks that are geographically proximate have an informational advantage in assessing local firms' R&D activities, they are better positioned to identify high-potential borrowers and provide financial support [47, 48]. The need to seek credit from more distant banking institutions could indicate that local lenders have a poor opinion regarding the operational status or innovative capabilities of the companies in question. In such cases, borrowing companies may have to accept less favorable loan terms. Moreover, the alienation associated with spatial separation can undermine the development of long-term, stable bank–borrower relationships that are crucial in supporting the step-by-step R&D process [49]. Hence, a contrasting hypothesis is also given:

**Hypothesis 3b**. Nonlocal loans have a negative effect on R&D investment.

## Interest rate liberalization policy

In June 2012, the Central Bank of China issued a policy to liberalize loan interest rates, adjusting the lower limit to 0.7 times its original value. This was followed by the complete removal of

the lower limit in July 2013, meaning financial institutions are allowed to independently determine loan rates. Benefiting from this policy change, companies have been able to access bank financing at lower costs [50]. The growth in disposable capital stimulates innovative behavior [51]; however, the higher pricing power of banks can exacerbate discrimination against nonlocal borrowers [52]. The information asymmetries associated with geographic distance can lead to higher information costs and default risks for nonlocal loans, prompting banks to impose tighter restrictions on factors such as repayment periods, interest rates, and loan amounts [12]. While the interest rate liberalization policy has enabled further financial reform, it may have widened the gap between local and nonlocal lending. Although nonlocal loans can supplement available capital, they may have also increased repayment pressures for borrowers since the implementation of the liberalization policy [52]. Therefore, the following hypothesis is proposed:

**Hypothesis 4**. The effect of nonlocal loans on R&D investment has become negative since the interest rate liberalization policy.

## Methods and data

### Data sources

This paper collects the annual financial data of A-share listed companies in China from 2007 to 2022. Since China implemented new accounting standards in 2007, the start year of the sample ensures the consistency of data measurement. All data is derived from the China Stock Market & Accounting Research Database, referring to the S1 Dataset. As this paper estimates the relationship between the interregional circulation of credit resources and company innovation, the sample retains listed companies with bank loans. Borrowing companies can be subsidiaries rather than listed companies, so the locations of borrowing companies, listed companies, and banks are obtained. While financial companies like banks and other financial institutions are subject to specific regulatory environments and accounting regulations [53], they are excluded from the sample. To avoid the influence of outliers, this study excludes any organization with missing data, initial public offerings, special treatment, or liability greater than its assets. All continuous variables are winsorized at 1% and 99%. The analysis is performed using Stata 14.0.

### R&D investments

This paper focuses on the effect of nonlocal loans on company innovation. The result of the ratio of R&D investments to revenue (RD) is widely used to measure the intensity of R&D investments [54], which can reflect the level of innovation enthusiasm. There can be a considerable time lag between funding acquisition and R&D activities, so the indicator for the next year (F.RD) is used to establish causality.

### Nonlocal loans

There is regional segmentation in the loan industry. Despite the increasing prevalence of the exchange of resources among different cities, banks still prefer to cooperate with companies in the same province [23]. Consequently, this study defines nonlocal loans using provincial boundaries instead of city boundaries. When making loan decisions, banks may take the asset scale, social reputation, repayment ability, and other factors of the listed company into consideration. Meanwhile, it is often the case that a subsidiary borrows money for the operations of

its parent company [55]. Hence, if the borrower, which may be the parent, subsidiary, or listed company, is not located in the same province as the bank, the loans are regarded as nonlocal (Nonlocal_d). The number of nonlocal loans is standardized with the year-end total assets (Nonlocal).

## Control variables

Financial constraint is a key factor in efforts to obtain credit resources [37]. Referring to a study conducted by Hadlock and Pierce [56], this paper utilizes a comparably exogenous index (SA) to measure financial constraints. This calculation only involves the size and age of companies, thus avoiding some endogenous problems. A lower value for the SA variable indicates that companies are confronted with more serious financial constraints.

Published in August 2020, the Innovation Index Report of Chinese Listed Companies in 2020 issued by Zhejiang University and Shenzhen Press Group clarifies that the index's top 500 companies are predominantly located in Beijing, Shenzhen, Shanghai, and Hangzhou. These cities have highly developed finance systems. The effect of finance on company innovation cannot be ignored, so the financial development level is considered in this study. Goldsmith [57] obtains the country's financial development level by calculating the ratio of the value of financial assets to gross domestic product (GDP). Referring to previous research [58], this paper measures the financial development level of the province (Fin) by calculating the ratio of the provincial balances of deposits and loans in financial institutions to provincial GDP.

According to innovation theory [59], innovation is spontaneous behavior whereby companies create a new combination of production factors to improve their profit margins. Following recent studies, this paper selects various firm-level control variables that may influence company innovation. Given their advantages in terms of capital and talent, large or mature firms are capable of innovation [60, 61]; thus, firm size (Size) and firm age (Age) are controlled for in this study. As innovation requires long-term steady capital input, it may be constrained by payment pressure from high leverage (Lev) [62] or be encouraged by a continuously positive operating cash flow (OCF) [55, 63]. Fan and Wang [64] and Fu et al. [65] argue that firms with state ownership (Gov) or high tangibility (PPE) are more likely to achieve loan approval and consequently embark on innovation activities. Furthermore, considering the decision-making power [66], this paper controls for the largest shareholder's ownership proportion (Top1). Detailed variable definitions are provided in Table 1.

## Model design

This paper utilizes the following six main regression models:

$$
\begin{aligned}
Nonlocal_{i,t} &= \alpha_0 + \alpha_1 Fin_{i,t} + \alpha_2 Size_{i,t} + \alpha_3 Lev_{i,t} + \alpha_4 Age_{i,t} + \alpha_5 Gov_{i,t} + \alpha_6 PPE_{i,t} \\
&+ \alpha_7 OCF_{i,t} + \alpha_8 TOP1_{i,t} + \alpha_9 Industry_i + \alpha_{10} Year_t + \varepsilon_{i,t}
\end{aligned}
\tag{1}
$$

$$
\begin{aligned}
Nonlocal_{i,t+1} &= \alpha_0 + \alpha_1 Fin_{i,t} + \alpha_2 Size_{i,t} + \alpha_3 Lev_{i,t} + \alpha_4 Age_{i,t} + \alpha_5 Gov_{i,t} + \alpha_6 PPE_{i,t} \\
&+ \alpha_7 OCF_{i,t} + \alpha_8 TOP1_{i,t} + \alpha_9 Industry_i + \alpha_{10} Year_t + \varepsilon_{i,t}
\end{aligned}
\tag{2}
$$

$$
\begin{aligned}
Nonlocal_{i,t} &= \alpha_0 + \alpha_1 SA_{i,t} + \alpha_2 Size_{i,t} + \alpha_3 Lev_{i,t} + \alpha_4 Age_{i,t} + \alpha_5 Gov_{i,t} + \alpha_6 PPE_{i,t} \\
&+ \alpha_7 OCF_{i,t} + \alpha_8 TOP1_{i,t} + \alpha_9 Industry_i + \alpha_{10} Year_t + \varepsilon_{i,t}
\end{aligned}
\tag{3}
$$

**Table 1. Variable definition.**

| Variable | Variable definition |
|---|---|
| RD | Ratio of R&D investment to revenue. |
| Nonlocal | Ratio of nonlocal loans (office address as the division basis) to year-end assets. |
| Fin | Ratio of the provincial balances of deposits and loans in financial institutions to provincial GDP. |
| SA | The detailed calculation could refer to Hadlock and Pierce (2010). |
| RepayPres | Ratio of interest expenses to year-end total assets. |
| Nonlocal_d | It would be 1 if the listed company gains nonlocal loans (office address as the division basis) and be 0 otherwise. |
| Hightech | Natural logarithm of number of high-tech firms that the listed company owns. |
| Lambda | Inverse mills ratio. |
| Size | Natural logarithm of year-end assets. |
| Lev | Ratio of year-end liability to year-end assets. |
| Age | Natural logarithm of number of years since the firm is listing. |
| Gov | It would be 1 if the listed company is state-owned and be 0 otherwise. |
| PPE | Ratio of year-end fixed assets to year-end assets. |
| OCF | Ratio of operating cash flow to year-end assets. |
| Top1 | The proportion of the largest shareholder. |
| Industry | Industry fixed effects. |
| Year | Year fixed effects. |

$$
\begin{aligned}
Nonlocal_{i,t+1} = {} & \alpha_0 + \alpha_1 SA_{i,t} + \alpha_2 Size_{i,t} + \alpha_3 Lev_{i,t} + \alpha_4 Age_{i,t} + \alpha_5 Gov_{i,t} + \alpha_6 PPE_{i,t} \\
& + \alpha_7 OCF_{i,t} + \alpha_8 TOP1_{i,t} + \alpha_9 Industry_i + \alpha_{10} Year_t + \varepsilon_{i,t}
\end{aligned}
\tag{4}
$$

$$
\begin{aligned}
RD_{i,t} = {} & \alpha_0 + \alpha_1 Nonlocal_{i,t} + \alpha_2 Fin_{i,t} + \alpha_3 SA_{i,t} + \alpha_4 Size_{i,t} + \alpha_5 Lev_{i,t} + \alpha_6 Age_{i,t} + \alpha_7 Gov_{i,t} \\
& + \alpha_8 PPE_{i,t} + \alpha_9 OCF_{i,t} + \alpha_{10} TOP1_{i,t} + \alpha_{11} Industry_i + \alpha_{12} Year_t + \varepsilon_{i,t}
\end{aligned}
\tag{5}
$$

$$
\begin{aligned}
RD_{i,t+1} = {} & \alpha_0 + \alpha_1 Nonlocal_{i,t} + \alpha_2 Fin_{i,t} + \alpha_3 SA_{i,t} + \alpha_4 Size_{i,t} + \alpha_5 Lev_{i,t} + \alpha_6 Age_{i,t} + \alpha_7 Gov_{i,t} \\
& + \alpha_8 PPE_{i,t} + \alpha_9 OCF_{i,t} + \alpha_{10} TOP1_{i,t} + \alpha_{11} Industry_i + \alpha_{12} Year_t + \varepsilon_{i,t}
\end{aligned}
\tag{6}
$$

Models (1) and (3) examine the cause of nonlocal loans; Model (5) is applied to assess the effect of the nonlocal loans on company innovation; and Models (2), (4), and (6) employ the value of the dependent variable for the next period to highlight causality rather than statistical association. Ordinary least squares regression can be accepted. All of the regressions control for the industry and year fixed effects, which substitute for unobservable factors that vary by industry and year level; however, their coefficients are too many to report and consequently have been omitted from the empirical results [53]. A robust standard error is applied to eliminate the possible interference of autocorrelation and heteroscedasticity in the significance of the coefficients.

## Results

### Descriptive results

Fig 1 reports the borrowing situation of Chinese companies from 2007 to 2022. Overall, credit resources have become increasingly important to company financing. The number of companies with loans experienced considerable growth in 2013. The interest rate liberalization policy

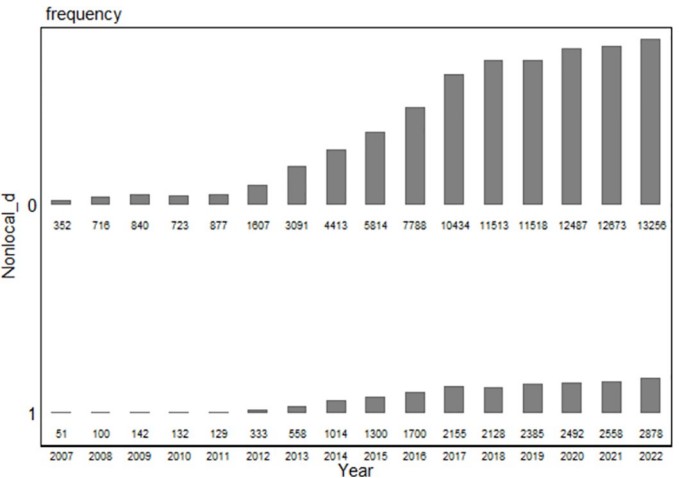

**Fig 1. Borrowing situation of Chinese companies from 2007 to 2022.**

has led to lower financing costs since 2013, meaning companies are more willing to participate in the lending industry. The shrinkage in the loan market from 2019 to 2021 can be explained by the economic downturn caused by the COVID-19 pandemic. With the economic recovery in China, the loan market once again expanded in 2022. The comparison results in Fig 1 indicate that companies are less likely to receive nonlocal loans. Although the policy has resulted in increased financial reforms, the market segmentation in the loan industry persists.

Table 2 tests the difference between local and nonlocal loans. For the full sample, the comparison results demonstrate that companies prefer to extend the repayment period and accept higher interest rates when signing agreements with nonlocal banks. The information asymmetry caused by long distances also encourages banks to charge more for risk compensation and issue loans on a smaller scale [41]. Considering the subsample size and loan approval interval, this paper takes 2013 as the dividing year of the interest rate liberalization policy. Compared to local loans, nonlocal loans were larger and had longer repayment periods before the policy implementation. These advantages have disappeared as banks have had more autonomy in the lending industry since 2013. It is noteworthy that distance is an important factor that affects loan pricing throughout the sample period. Companies are under higher repayment pressure for nonlocal loans.

**Table 2. Comparison between local and nonlocal loans.**

| Sample | Characteristic | Local loans | Nonlocal loans | Difference | p-value |
|---|---|---|---|---|---|
| The full sample | Repayment period (Years) | 1.660 | 1.831 | -0.171 | 0.093 |
| | Interest rate (%) | 5.984 | 6.824 | -0.840 | 0.000 |
| | Scale (Million yuan) | 14413.050 | 11768.826 | 2644.224 | 0.074 |
| Before 2013 | Repayment period (Years) | 1.605 | 2.192 | -0.587 | 0.000 |
| | Interest rate (%) | 6.458 | 7.361 | -0.903 | 0.000 |
| | Scale (Million yuan) | 2120.733 | 2837.785 | -717.052 | 0.000 |
| After 2013 | Repayment period (Years) | 1.670 | 1.773 | -0.103 | 0.389 |
| | Interest rate (%) | 5.658 | 6.639 | -0.981 | 0.000 |
| | Scale (Million yuan) | 16175.440 | 12826.746 | 3348.694 | 0.045 |

**Table 3. The effect of the interest rate liberalization policy on loans.**

| Attribute | Characteristic | Before 2013 | After 2013 | Difference | p-value |
|---|---|---|---|---|---|
| Local loans | Repayment period (Years) | 1.605 | 1.670 | -0.065 | 0.573 |
| | Interest rate (%) | 6.458 | 5.658 | 0.800 | 0.000 |
| | Scale (Million yuan) | 2120.733 | 16175.440 | -14054.707 | 0.000 |
| Nonlocal loans | Repayment period (Years) | 2.192 | 1.773 | 0.419 | 0.000 |
| | Interest rate (%) | 7.361 | 6.639 | 0.722 | 0.001 |
| | Scale (Million yuan) | 2837.785 | 12826.746 | -9988.961 | 0.000 |

Table 3 reports the effect of the interest rate liberalization policy on loans. Both local and nonlocal loans have experienced a decrease in interest rates and an increase in loan scale since 2013. The policy has resulted in lower repayment pressure and has encouraged companies to sign a larger volume of loans, thus promoting the development of the credit industry.

Table 4 provides the descriptive results of the study sample. The mean value (0.044) of Nonlocal implies that companies do not obtain nonlocal loans on a regular basis. However, the maximum value of Nonlocal is 0.860, meaning that some companies are still highly reliant on nonlocal loans for external financing. The mean value and standard deviation of RD are 0.044 and 0.043, respectively. The innovation enthusiasm of companies varies. The standard deviation (1.543) of Fin demonstrates that there are substantial differences in the level of financial development among provinces. The standard deviation of SA (0.233) implies that the differences in financial constraints among listed companies are comparatively small, while the results of Size and Age demonstrate that the companies in the sample are generally large and mature. The minimum (0.079) and maximum (0.891) of Lev imply different operating strategies; some companies prefer conservative operations and maintain solvency, while others are willing to undertake higher financial risk. Similarly, the results of PPE reflect various levels of asset liquidity, while the OCF results show diverse profitability of the main business of the companies. The mean (0.314) of Gov indicates that most listed companies are not state-owned. Finally, the mean (0.330) of Top1 shows that the equity concentration of listed companies is comparably high. The results for the control variables are roughly consistent with recent findings [15, 29, 67].

**Table 4. Descriptive results of the variables.**

| Variable | Obs | Mean | Std.Dev. | Min | Max |
|---|---|---|---|---|---|
| Nonlocal | 8089 | 0.044 | 0.125 | 0.000 | 0.860 |
| RD | 8089 | 0.044 | 0.043 | 0.000 | 0.246 |
| Fin | 8089 | 3.707 | 1.543 | 1.506 | 7.575 |
| SA | 8089 | -3.784 | 0.233 | -4.408 | -3.252 |
| Size | 8089 | 22.191 | 1.143 | 20.128 | 25.554 |
| Lev | 8089 | 0.443 | 0.192 | 0.079 | 0.891 |
| Age | 8089 | 2.844 | 0.342 | 1.099 | 3.807 |
| Gov | 8089 | 0.314 | 0.464 | 0.000 | 1.000 |
| PPE | 8089 | 0.206 | 0.145 | 0.004 | 0.651 |
| OCF | 8089 | 0.038 | 0.063 | -0.146 | 0.214 |
| Top1 | 8089 | 0.330 | 0.143 | 0.081 | 0.708 |

**Table 5. The causes of nonlocal loans.**

| | (1) | (2) | (3) | (4) |
|---|---|---|---|---|
| | **Nonlocal** | **F.Nonlocal** | **Nonlocal** | **F.Nonlocal** |
| Fin | -0.006*** | -0.006*** | | |
| | (-6.891) | (-6.229) | | |
| SA | | | -0.018 | -0.019 |
| | | | (-1.420) | (-1.109) |
| Size | -0.002 | 0.001 | -0.002 | -0.000 |
| | (-1.271) | (0.331) | (-1.500) | (-0.095) |
| Lev | 0.073*** | 0.075*** | 0.076*** | 0.079*** |
| | (7.317) | (5.843) | (7.972) | (6.196) |
| Age | -0.002 | 0.001 | -0.013 | -0.021* |
| | (-0.430) | (0.260) | (-1.537) | (-1.863) |
| Gov | 0.027*** | 0.031*** | 0.026*** | 0.030*** |
| | (7.052) | (6.277) | (7.302) | (6.038) |
| PPE | -0.040*** | -0.046*** | -0.026** | -0.032** |
| | (-3.411) | (-2.937) | (-2.348) | (-2.081) |
| OCF | -0.004 | 0.014 | -0.002 | 0.013 |
| | (-0.163) | (0.453) | (-0.078) | (0.423) |
| Top1 | 0.007 | 0.007 | 0.009 | 0.008 |
| | (0.746) | (0.610) | (1.055) | (0.622) |
| _cons | 0.067* | 0.000 | 0.002 | -0.089 |
| | (1.846) | (0.004) | (0.042) | (-1.409) |
| Industry | Yes | Yes | Yes | Yes |
| Year | Yes | Yes | Yes | Yes |
| N | 8089 | 6085 | 8089 | 6128 |
| Adj. $R^2$ | 0.041 | 0.044 | 0.036 | 0.041 |

Note: The t statistics are shown in parentheses.

*, **, and *** indicate that coefficients are respectively significant at the levels of 10%, 5%, and 1%.

## Baseline results

This study explores the causes of nonlocal loans in terms of financial restraint and the abundance of local credit resources. Column (1) in Table 5 indicates that companies seek more nonlocal loans when the financial development level of their province is low. Thus, Hypothesis 1 is verified. From the demand perspective [68], companies are active in pursuing nonlocal loans when the credit resources in their province are scarce. Nonlocal loans play an essential role in supplementing capital. Column (2) in Table 5 shows an insignificant relationship between financial constraints and nonlocal loans, meaning that Hypothesis 2 is rejected. Financial constraints make it difficult for companies to obtain financial support from both local and nonlocal banks [37]. Such companies do not receive much support from nonlocal banks, meaning cross-provincial loans are unlikely to be granted.

Column (1) in Table 6 indicates that nonlocal loans have a significantly positive effect on R&D investment intensity. The causality is confirmed by Column (2) as the R&D investment intensity continues to grow in the next period. Hypothesis 3a is therefore supported. The injection of nonlocal loans increases the start-up capital for innovation, and supplementary capital provided via these loans serves to constantly reinforce the innovation preference of companies. Similarly, the importance of available capital to innovation is verified by the Fin and SA results.

**Table 6. The effect of nonlocal loans on company innovation.**

| | The full sample | | Before 2013 | | After 2013 | |
|---|---|---|---|---|---|---|
| | (1) | (2) | (3) | (4) | (5) | (6) |
| | RD | F.RD | RD | F.RD | RD | F.RD |
| Nonlocal | 0.007** | 0.003*** | -0.010 | -0.000 | 0.009*** | 0.003*** |
| | (2.456) | (2.693) | (-1.335) | (-0.014) | (2.654) | (2.739) |
| Fin | 0.002*** | 0.002*** | 0.002*** | 0.001 | 0.002*** | 0.002*** |
| | (6.207) | (4.257) | (2.953) | (0.678) | (5.447) | (4.220) |
| SA | 0.017*** | 0.019*** | 0.014 | 0.011 | 0.021*** | 0.024*** |
| | (3.939) | (3.191) | (1.295) | (0.810) | (4.378) | (3.497) |
| Size | -0.002*** | -0.002*** | -0.003*** | -0.001 | -0.002*** | -0.002*** |
| | (-5.146) | (-2.855) | (-2.665) | (-0.737) | (-4.729) | (-2.989) |
| Lev | -0.039*** | -0.035*** | -0.048*** | -0.043*** | -0.037*** | -0.033*** |
| | (-12.806) | (-9.358) | (-7.315) | (-5.765) | (-10.727) | (-7.622) |
| Age | 0.000 | 0.004 | -0.004 | -0.003 | 0.004 | 0.008* |
| | (0.163) | (0.929) | (-0.601) | (-0.424) | (1.237) | (1.699) |
| Gov | -0.003*** | -0.004*** | -0.000 | -0.000 | -0.004*** | -0.005*** |
| | (-3.430) | (-3.333) | (-0.110) | (-0.111) | (-3.629) | (-3.562) |
| PPE | -0.026*** | -0.032*** | -0.014* | -0.028*** | -0.029*** | -0.033*** |
| | (-8.014) | (-7.948) | (-1.864) | (-3.534) | (-7.890) | (-7.201) |
| OCF | -0.033*** | -0.030*** | -0.009 | 0.012 | -0.038*** | -0.040*** |
| | (-4.444) | (-3.238) | (-0.647) | (0.785) | (-4.397) | (-3.622) |
| Top1 | -0.020*** | -0.021*** | -0.005 | -0.006 | -0.024*** | -0.025*** |
| | (-7.192) | (-5.955) | (-0.875) | (-0.797) | (-7.505) | (-6.047) |
| _cons | 0.142*** | 0.137*** | 0.139*** | 0.099*** | 0.165*** | 0.164*** |
| | (9.852) | (7.046) | (4.190) | (2.614) | (10.200) | (7.421) |
| Industry | Yes | Yes | Yes | Yes | Yes | Yes |
| Year | Yes | Yes | Yes | Yes | Yes | Yes |
| N | 8089 | 5021 | 1403 | 970 | 6686 | 4051 |
| Adj. R$^2$ | 0.299 | 0.277 | 0.331 | 0.264 | 0.288 | 0.272 |

Note: The t statistics are shown in parentheses.

*, **, and *** indicate that coefficients are respectively significant at the levels of 10%, 5%, and 1%.

Companies that are confronted with lower financial restraints or possess more financial resources prefer to add innovation investment.

From the heterogeneous analysis, the positive effect of nonlocal loans is only apparent in the period following 2013. Although nonlocal loans do not have an advantage in terms of interest rate or scale over local loans in the post-2013 period, the implementation of the interest rate liberalization policy has had a positive influence on the nonlocal lending industry. Consequently, Hypothesis 4 is rejected. The positive effect of nonlocal loans on company innovation might be attributed to falling interest rates or capital replenishment, which are analyzed in the discussion section.

## Robustness tests

### Reciprocal causation test

Innovative companies are usually involved in many R&D projects that entail a high demand for financing. Such companies may seek multiple loans from different financial institutions to

**Table 7. Reciprocal causation test.**

| | (1) | (2) | (3) | (4) |
|---|---|---|---|---|
| | Nonlocal_d | F.Nonlocal_d | Nonlocal_d | F.Nonlocal_d |
| RD | 0.655 | -0.138 | | |
| | (0.881) | (-0.163) | | |
| Hightech | | | 0.045 | -0.028 |
| | | | (1.105) | (-0.612) |
| Fin | -0.113*** | -0.083*** | -0.105*** | -0.079*** |
| | (-5.945) | (-4.060) | (-5.007) | (-3.564) |
| SA | -0.339 | -0.560* | -0.230 | -0.421 |
| | (-1.181) | (-1.668) | (-0.678) | (-1.071) |
| Size | 0.299*** | 0.368*** | 0.303*** | 0.396*** |
| | (10.519) | (11.132) | (8.782) | (9.998) |
| Lev | 2.056*** | 1.574*** | 2.095*** | 1.740*** |
| | (12.144) | (8.086) | (11.096) | (8.140) |
| Age | -0.184 | -0.229 | -0.085 | -0.067 |
| | (-0.910) | (-0.979) | (-0.363) | (-0.246) |
| Gov | 0.287*** | 0.135* | 0.292*** | 0.144* |
| | (4.688) | (1.948) | (4.282) | (1.878) |
| PPE | 0.136 | 0.079 | 0.273 | -0.022 |
| | (0.657) | (0.334) | (1.132) | (-0.082) |
| OCF | -0.910** | -0.545 | -1.293*** | -0.958* |
| | (-2.111) | (-1.093) | (-2.656) | (-1.718) |
| Top1 | -0.331* | -0.457** | -0.464** | -0.630*** |
| | (-1.772) | (-2.176) | (-2.202) | (-2.690) |
| _cons | -9.288*** | -10.651*** | -9.403*** | -11.222*** |
| | (-9.601) | (-9.455) | (-8.382) | (-8.625) |
| Industry | Yes | Yes | Yes | Yes |
| Year | Yes | Yes | Yes | Yes |
| N | 8089 | 6083 | 6738 | 5150 |
| Pseudo $R^2$ | 0.090 | 0.084 | 0.095 | 0.091 |

Note: The t statistics are shown in parentheses.

*, **, and *** indicate that coefficients are respectively significant at the levels of 10%, 5%, and 1%.

get sufficient capital [68]. Hence, there might be a reciprocal causation between nonlocal loans and R&D investment intensity.

A logistic regression is conducted to determine whether the demand for innovation increases the possibility of the acquisition of nonlocal loans. The dummy variable Nonlocal_d is 1 if the listed company receives nonlocal loans, and 0 otherwise. As well as R&D investment, this study has collected the number of high-tech firms that belong to each listed company. The natural logarithm of its value (Hightech) has been estimated to present the innovation demand of the listed company. Columns (1) to (4) in Table 7 demonstrate that significant innovation demand does not motivate companies to pursue nonlocal loans. Thus, there is no significant reciprocal causation.

## Selection bias test

The sample for this paper is composed of listed companies that have received loans; however, loans are not granted at random. The scale, payment ability, and operating conditions of listed

companies are observed before banks decide whether loans are granted. The requirements may be stricter if the bank grants nonlocal loans, meaning there may be a selection bias in the sample. This section applies the two-stage selection model from Heckman [69] to re-estimate the relationship between the nonlocal loans and R&D investment intensity.

$$Probit\left(NonlocaLd_{1_{i,t}}\right) = \sum \alpha * Control_{i,t} + \alpha_1 Industry_i + \alpha_2 Year_t + \varepsilon_{i,t} \tag{7}$$

$$Lambda_{i,t} = \varphi\left(\sum \widehat{\alpha} * Factors_{i,t}\right) / \phi\left(\sum \widehat{\alpha} * Factors_{i,t}\right) \tag{8}$$

$$RD_{i,t} = \alpha_0 + \alpha_1 Nonlocal_{i,t} + \alpha_2 Lambda_{i,t} + \sum \alpha * Control_{i,t} + \alpha_3 Industry_i + \alpha_4 Year_t + \varepsilon_{i,t} \tag{9}$$

$$RD_{i,t+1} = \alpha_0 + \alpha_1 Nonlocal_{i,t} + \alpha_2 Lambda_{i,t} + \sum \alpha * Control_{i,t} + \alpha_3 Industry_i + \alpha_4 Year_t + \varepsilon_{i,t} \tag{10}$$

The first stage, Model (7), investigates whether the agreement of nonlocal loans is affected by firm-level, industry-level, or year-level factors. $\sum \alpha * Control_{i,t}$ is the abbreviation of all the firm-level control variables. All of the explanatory variables in Model (7) can be described as heterogeneity factors (Factors). The estimated coefficients of the Factors in the first stage are collected. In Model (8), the inverse Mills ratio (Lambda) is calculated using the density function $\varphi(\cdot)$ and cumulative distribution function $\phi(\cdot)$. Lambda is linearly correlated with the estimated error in Model (7). This is then applied to the second stage in Models (9) and (10). If its coefficient (Lambda) is significantly different from 0, there is a selection bias in the sample. Then, the Nonlocal coefficient in the second stage would be more persuasive than it is in previous empirical results.

Columns (1) and (2) in Table 8 show that the Lambda coefficient is insignificant, meaning there is no significant selection bias. The estimated results in the second stage also demonstrate that nonlocal loans encourage companies to increase the intensity of R&D investment.

Additionally, the propensity score matching (PSM) method of caliper 0.01 is used. Companies with nonlocal loans can be regarded as the treatment group and others as the control group. Two groups can be matched with similar characteristics in terms of control variables through PSM, which can eliminate the effects of sample selection bias. After removing the unmatched sample, the remaining sample is re-estimated. Columns (3) and (4) in Table 8 indicate that the main findings in this paper are consistent.

## Alternative estimation

Different measurements of nonlocal loans may affect the significance of the empirical results. For example, a study by Aivazian et al. [70] obtains the debt maturity structure by calculating the proportion of long-term debt in total debt. This study standardizes the nonlocal loans by the total loans (Nonlocal_r) for the robustness check. Alternatively, the registered address can be used as a new division basis to define nonlocal loans. When the registered address of the borrowing company or listed company is not in the same province as the bank, the loans are regarded as nonlocal. The proportion (Nonlocal2) of nonlocal loans to year-end assets can then be estimated. Table 9 reports the regression results with alternative variables, confirming the positive effect of nonlocal loans on R&D investment. These results are consistent with previous findings in this paper.

**Table 8. Selection bias test.**

|  | (1) | (2) | (3) | (4) |
|---|---|---|---|---|
|  | RD | F.RD | RD | F.RD |
| Nonlocal | 0.008** | 0.007** | 0.007** | 0.006** |
|  | (2.350) | (1.980) | (2.401) | (2.001) |
| Lambda | -0.000 | -0.001 |  |  |
|  | (-0.235) | (-0.842) |  |  |
| Fin | 0.002*** | 0.002*** | 0.002*** | 0.002*** |
|  | (6.199) | (5.164) | (6.172) | (5.247) |
| SA | 0.017*** | 0.020*** | 0.015*** | 0.017*** |
|  | (3.940) | (3.799) | (3.261) | (3.012) |
| Size | -0.002*** | -0.002*** | -0.002*** | -0.002*** |
|  | (-5.146) | (-3.296) | (-4.869) | (-3.175) |
| Lev | -0.039*** | -0.037*** | -0.039*** | -0.037*** |
|  | (-12.837) | (-10.679) | (-12.504) | (-10.459) |
| Age | 0.001 | 0.004 | -0.000 | 0.003 |
|  | (0.165) | (1.160) | (-0.116) | (0.770) |
| Gov | -0.003*** | -0.005*** | -0.003*** | -0.004*** |
|  | (-3.439) | (-4.116) | (-3.255) | (-3.844) |
| PPE | -0.026*** | -0.034*** | -0.026*** | -0.034*** |
|  | (-7.996) | (-8.836) | (-7.827) | (-8.542) |
| OCF | -0.033*** | -0.030*** | -0.035*** | -0.033*** |
|  | (-4.444) | (-3.402) | (-4.573) | (-3.664) |
| Top1 | -0.020*** | -0.022*** | -0.021*** | -0.022*** |
|  | (-7.193) | (-6.702) | (-7.294) | (-6.499) |
| _cons | 0.142*** | 0.141*** | 0.138*** | 0.141*** |
|  | (9.852) | (7.970) | (9.095) | (7.453) |
| Industry | Yes | Yes | Yes | Yes |
| Year | Yes | Yes | Yes | Yes |
| N | 8089 | 6083 | 7809 | 5854 |
| Adj. $R^2$ | 0.299 | 0.281 | 0.292 | 0.272 |

Note: The t statistics are shown in parentheses.

*, **, and *** indicate that coefficients are respectively significant at the levels of 10%, 5%, and 1%.

## Discussion

The existing research widely criticizes nonlocal loans because of their drawbacks but does not explain these issues [11, 12]. This study acknowledges the disadvantages of nonlocal loans in terms of repayment period, interest rate, and scale. The results highlight the function of nonlocal loans as supplementary capital, especially in financially undeveloped areas. As credit resources are unevenly distributed across regions, some regions may become stuck in a cycle of limited credit availability, low investment, and stagnant economic development [71]. This article proposes a possible solution. The interregional circulation of credit resources can alleviate credit traps and promote more balanced and inclusive regional development. The process can be accelerated via policy interventions, interregional cooperation and integration, improvement in information transparency, and the development of financial services [72, 73].

This article indicates that an increase in disposable capital can enhance the willingness of companies to innovate. The findings are roughly consistent with existing research, which

**Table 9. Alternative estimation of nonlocal loans.**

|  | (1) | (2) | (3) | (4) |
|---|---|---|---|---|
|  | RD | F.RD | RD | F.RD |
| Nonlocal_r | 0.004*** | 0.004** |  |  |
|  | (3.090) | (2.395) |  |  |
| Nonlocal2 |  |  | 0.007** | 0.006** |
|  |  |  | (2.433) | (1.985) |
| Fin | 0.002*** | 0.002*** | 0.002*** | 0.002*** |
|  | (6.395) | (5.268) | (6.166) | (5.109) |
| SA | 0.016*** | 0.020*** | 0.017*** | 0.020*** |
|  | (3.846) | (3.748) | (3.943) | (3.800) |
| Size | -0.002*** | -0.002*** | -0.002*** | -0.002*** |
|  | (-5.317) | (-3.413) | (-5.172) | (-3.327) |
| Lev | -0.039*** | -0.037*** | -0.039*** | -0.036*** |
|  | (-12.812) | (-10.655) | (-12.824) | (-10.644) |
| Age | 0.000 | 0.004 | 0.000 | 0.004 |
|  | (0.077) | (1.111) | (0.160) | (1.154) |
| Gov | -0.003*** | -0.005*** | -0.003*** | -0.004*** |
|  | (-3.567) | (-4.196) | (-3.414) | (-4.056) |
| PPE | -0.027*** | -0.034*** | -0.026*** | -0.034*** |
|  | (-8.079) | (-8.852) | (-8.025) | (-8.857) |
| OCF | -0.033*** | -0.030*** | -0.033*** | -0.030*** |
|  | (-4.416) | (-3.391) | (-4.436) | (-3.401) |
| Top1 | -0.020*** | -0.022*** | -0.020*** | -0.022*** |
|  | (-7.161) | (-6.683) | (-7.194) | (-6.701) |
| _cons | 0.143*** | 0.141*** | 0.142*** | 0.141*** |
|  | (9.906) | (8.009) | (9.873) | (7.993) |
| Industry | Yes | Yes | Yes | Yes |
| Year | Yes | Yes | Yes | Yes |
| N | 8089 | 6085 | 8089 | 6085 |
| Adj. R$^2$ | 0.300 | 0.281 | 0.299 | 0.281 |

Note: The t statistics are shown in parentheses.

*, **, and *** indicate that coefficients are respectively significant at the levels of 10%, 5%, and 1%.

highlights the positive effect of credit resources on company innovation. However, some studies have demonstrated that companies in areas with sufficient credit obtain fewer nonlocal loans but have higher innovation output [58]. Notably, this article does not emphasize that nonlocal loans have more advantages compared to local loans but rather confirms the financing ability of the companies involved and the use of nonlocal loans to supplement disposable capital. Some literature provides similar viewpoints that the circulation of credit resources may facilitate the adoption and diffusion of new technologies [74]. In the remainder of this discussion, the causality is strengthened by exploring the effective paths of nonlocal loans and observing the results of R&D manipulation and R&D output.

## Effective paths

Based on the main empirical results, this section discusses the effective paths of nonlocal loans in relation to company innovation. As companies can obtain multiple loans within one year, it

**Table 10. The effective paths of nonlocal loans on company innovation.**

| | (1) | (2) | (3) |
|---|---|---|---|
| | RD | RD | RD |
| RepayPres*Nonlocal | -34.557 | | |
| | (-1.366) | | |
| Fin*Nonlocal | | -0.011*** | |
| | | (-3.008) | |
| SA*Nonlocal | | | -0.010 |
| | | | (-0.710) |
| Nonlocal | 0.008** | 0.042*** | -0.031 |
| | (2.505) | (3.722) | (-0.572) |
| RepayPres | -3.303** | | |
| | (-2.228) | | |
| Fin | 0.002*** | 0.002*** | 0.002*** |
| | (6.191) | (6.795) | (6.190) |
| SA | 0.017*** | 0.017*** | 0.017*** |
| | (3.962) | (3.908) | (4.039) |
| Size | -0.002*** | -0.002*** | -0.002*** |
| | (-5.063) | (-5.145) | (-5.137) |
| Lev | -0.039*** | -0.039*** | -0.039*** |
| | (-12.803) | (-12.716) | (-12.805) |
| Age | 0.001 | 0.001 | 0.001 |
| | (0.165) | (0.180) | (0.195) |
| Gov | -0.003*** | -0.003*** | -0.003*** |
| | (-3.457) | (-3.623) | (-3.448) |
| PPE | -0.026*** | -0.026*** | -0.026*** |
| | (-8.041) | (-7.976) | (-8.012) |
| OCF | -0.033*** | -0.033*** | -0.033*** |
| | (-4.492) | (-4.417) | (-4.450) |
| Top1 | -0.020*** | -0.021*** | -0.020*** |
| | (-7.206) | (-7.252) | (-7.189) |
| _cons | 0.142*** | 0.140*** | 0.144*** |
| | (9.827) | (9.692) | (9.860) |
| Industry | Yes | Yes | Yes |
| Year | Yes | Yes | Yes |
| N | 8089 | 8089 | 8089 |
| Adj. $R^2$ | 0.299 | 0.301 | 0.299 |

Note: The t statistics are shown in parentheses.

*, **, and *** indicate that coefficients are respectively significant at the levels of 10%, 5%, and 1%.

is more appropriate to measure repayment pressure (RepayPres) by observing interest expenses rather than interest rates [75]. The interest expenses can be standardized with the year-end total assets. Column (1) in Table 10 shows that repayment pressure can hinder company innovation; meanwhile, nonlocal loans do not significantly alleviate the negative connection between repayment pressure and company innovation. This is understandable since nonlocal loans produce more repayment pressure than local loans. Column (2) in Table 10 highlights that nonlocal loans can alleviate the strong positive relationship between the financial development level of a province and the R&D investment intensity of companies located

in that province. The R&D projects of companies in financially developed areas are not especially reliant on nonlocal loans, while the provision of nonlocal loans can satisfy the innovation demand of companies in financially undeveloped areas. As indicated by Column (3) in Table 10, nonlocal loans do not encourage companies with serious financial constraints to increase R&D investment. Due to a shortage of available capital, companies with financing restraints may prioritize operating activities over innovation.

Polarization effect theory refers to a positive feedback phenomenon whereby developed regions absorb economic elements [76]. Without external intervention, the polarization effect dominates the market and triggers the stratification of the economy. The allocation of credit resources to more financially developed provinces can be reasonably seen as a better use of capital. However, interregional circulation can remedy the lack of local credit resources as these capital replenishments can stimulate companies in financially undeveloped areas to increase their R&D investment, which in turn helps to relieve imbalances in regional economic development.

### R&D manipulation

It should be noted that the mean value of RD (0.044) is within the 3–5% interval specified in the Identification and Management Methods of High-Tech Companies in China (IMMHCC). Once companies are identified, they benefit from tax incentives. Therefore, an increase in R&D investment may be an opportunistic strategy to meet the policy requirements rather than an active R&D strategy. Such behavior, which leads to minimal innovation output and lowers the efficiency of capital usage, is known as R&D manipulation [77]. Hull et al. [78] demonstrate that R&D manipulation increases the number of utility or design patents but not the number of invention patents. As a result, companies' high-quality development is inhibited, and their competitive advantage is reduced [79].

Based on the findings of Zhang et al. [77], R&D manipulation can be recognized if the R&D investment intensity is less than 0.5% or 1% above the IMMHCC threshold. Thus, the dummy variable (Manipul1) is 1 if the excess value is less than 0.5%, and 0 otherwise. Similarly, an alternative variable (Manipul2) is set with an excess of 1%. Table 11 shows the logit regression results. The insignificant relationship between nonlocal loans and R&D manipulation illustrates that the increase in R&D investment is not an opportunistic behavior for tax incentives, implying that the infusion of nonlocal loans is used to undertake innovation.

### R&D output

An increased level of innovation output can be achieved if companies endeavor to innovate. The relationship between nonlocal loans and innovation output is expected to be positive. Some researchers have measured innovation output using patent applications and grants [16]; however, it is necessary to distinguish invention patents from others. Wen et al. [29] consider invention patents as symbols of technological innovation that directly promote technological development. Compared with invention patents, patents for utility models and design are easier to realize. They sometimes play a role in strategic innovation by helping companies gain tax incentives while contributing little to the competitive edge [17].

Consistent with the approach of Wen et al. [29], this research has collected the application numbers of all patents (Patents), invention patents (Invention), and utility model and design patents (Utility&Design). The data are logarithmically processed. Columns (1) and (2) in Table 12 provide evidence that nonlocal loans have a positive effect on innovation output, while Columns (3) to (6) indicate that the replenishment of funds strengthens the preference for high-quality innovation.

**Table 11. Nonlocal loans and R&D manipulation.**

|  | (1) | (2) | (3) | (4) |
|---|---|---|---|---|
|  | Manipul1 | F.Manipul1 | Manipul2 | F.Manipul2 |
| Nonlocal | 0.528 | 0.861 | 0.573 | 0.904 |
|  | (1.487) | (1.568) | (1.276) | (1.534) |
| Fin | -0.033 | -0.028 | -0.071*** | -0.058** |
|  | (-1.253) | (-0.915) | (-3.372) | (-2.417) |
| SA | -1.257** | -0.923 | -1.008*** | -0.884** |
|  | (-2.459) | (-1.473) | (-2.709) | (-1.961) |
| Size | -0.217*** | -0.213*** | -0.192*** | -0.220*** |
|  | (-4.677) | (-3.720) | (-5.414) | (-5.162) |
| Lev | 0.502** | 0.577** | 0.505*** | 0.670*** |
|  | (2.143) | (1.994) | (2.729) | (3.009) |
| Age | -0.998*** | -0.628 | -0.920*** | -0.785** |
|  | (-2.889) | (-1.436) | (-3.665) | (-2.561) |
| Gov | -0.072 | -0.028 | -0.110 | -0.018 |
|  | (-0.742) | (-0.244) | (-1.450) | (-0.202) |
| PPE | 0.826*** | 0.948** | 0.755*** | 0.565** |
|  | (2.771) | (2.566) | (3.203) | (1.985) |
| OCF | 1.772*** | 1.174 | 2.052*** | 2.112*** |
|  | (2.891) | (1.547) | (4.227) | (3.598) |
| Top1 | 0.562** | 0.957*** | 0.535** | 0.689*** |
|  | (2.112) | (3.012) | (2.500) | (2.736) |
| _cons | -0.905 | 0.043 | -0.379 | 1.109 |
|  | (-0.669) | (0.025) | (-0.352) | (0.852) |
| Industry | Yes | Yes | Yes | Yes |
| Year | Yes | Yes | Yes | Yes |
| N | 8089 | 6013 | 8089 | 6069 |
| Adj. $R^2$ | 0.056 | 0.060 | 0.066 | 0.061 |

Note: The t statistics are shown in parentheses.

*, **, and *** indicate that coefficients are respectively significant at the levels of 10%, 5%, and 1%.

## Conclusions and implications

This study divides loans into local and nonlocal ones in accordance with provincial boundaries. The findings indicate that companies are compelled to seek nonlocal loans when the provincial financial development level is comparably low. When the lack of local credit resources is eased, the supplementary funds provided by nonlocal loans encourage companies to increase R&D investment. Moreover, although company innovation in financially developed areas is not overly reliant on nonlocal loans, the injection of these loans can have a significant positive effect on the level of innovation enthusiasm in financially undeveloped areas. This study also excludes the possibility of R&D manipulation and highlights the increase in invention patent output. Interregional circulation may address the unbalanced distribution of regional credit resources and promote the development of financially undeveloped areas.

Based on the characteristics of nonlocal loans, this research provides some implications. A substantial geographic distance increases the difficulty in evaluating the loan default risk of the borrowing companies. Closer cooperation should be built for banks in different regions. Sharing regulatory information may be an efficient and rigorous way to identify high-quality

**Table 12. Nonlocal loans and R&D output.**

| | (1) | (2) | (3) | (4) | (5) | (6) |
|---|---|---|---|---|---|---|
| | **Patents** | **F.Patents** | **Invention** | **F.Invention** | **Utility&Design** | **F.Utility&Design** |
| Nonlocal | 0.465*** | 0.135** | 0.411** | 0.189** | 0.252 | -0.081 |
| | (3.156) | (1.988) | (2.548) | (2.250) | (1.481) | (-1.075) |
| Fin | 0.017 | 0.027* | 0.042*** | 0.056*** | 0.000 | -0.002 |
| | (1.586) | (1.898) | (3.560) | (3.622) | (0.039) | (-0.121) |
| SA | 0.182 | 0.219 | 0.274 | 0.162 | -0.085 | -0.061 |
| | (0.876) | (0.754) | (1.227) | (0.504) | (-0.424) | (-0.227) |
| Size | 0.570*** | 0.544*** | 0.625*** | 0.607*** | 0.227*** | 0.236*** |
| | (28.321) | (19.804) | (28.939) | (20.293) | (11.432) | (9.000) |
| Lev | -0.088 | -0.062 | -0.471*** | -0.477*** | -0.156 | -0.255* |
| | (-0.832) | (-0.438) | (-4.175) | (-3.208) | (-1.538) | (-1.928) |
| Age | -0.179 | -0.193 | -0.090 | -0.198 | 0.005 | 0.040 |
| | (-1.358) | (-1.094) | (-0.642) | (-1.013) | (0.040) | (0.250) |
| Gov | -0.086** | -0.172*** | -0.012 | -0.112** | -0.198*** | -0.188*** |
| | (-2.148) | (-3.281) | (-0.270) | (-1.990) | (-5.209) | (-3.876) |
| PPE | -1.548*** | -1.751*** | -1.296*** | -1.644*** | -1.553*** | -1.788*** |
| | (-11.366) | (-9.853) | (-9.241) | (-8.661) | (-11.911) | (-10.753) |
| OCF | 0.535* | 1.255*** | 0.666** | 1.360*** | 0.602** | 1.313*** |
| | (1.908) | (3.478) | (2.268) | (3.743) | (2.176) | (3.862) |
| Top1 | -0.211* | -0.421*** | -0.443*** | -0.613*** | 0.290** | 0.219 |
| | (-1.779) | (-2.738) | (-3.487) | (-3.762) | (2.448) | (1.464) |
| _cons | -8.924*** | -8.170*** | -10.429*** | -10.148*** | -4.238*** | -4.114*** |
| | (-15.195) | (-10.220) | (-16.786) | (-11.757) | (-7.418) | (-5.395) |
| Industry | Yes | Yes | Yes | Yes | Yes | Yes |
| Year | Yes | Yes | Yes | Yes | Yes | Yes |
| N | 4846 | 2864 | 4846 | 2864 | 4846 | 2864 |
| Adj. $R^2$ | 0.277 | 0.265 | 0.256 | 0.249 | 0.093 | 0.108 |

Note: The t statistics are shown in parentheses.

*, **, and *** indicate that coefficients are respectively significant at the levels of 10%, 5%, and 1%.

companies in financially undeveloped areas, helping banks provide financial support in an effective and targeted manner. In addition, the collaboration between banking institutions and the China National Intellectual Property Administration could be improved; this would make it easier for banks to recognize innovative companies with great potential in financially undeveloped areas.

This study contributes to the literature in several ways. Firstly, companies seeking nonlocal loans have been widely criticized due to the difficulties associated with monitoring their activities and high interest rates, yet there have been few explanations regarding the occurrence of nonlocal loans. Based on resource dependence theory, this paper explains why companies secure nonlocal loans and emphasizes the supplementary function of these loans in areas with insufficient local credit resources. Secondly, this paper enriches and extends the understanding of the economic consequences of loans with regard to company innovation, taking regional segmentation into consideration. Thirdly, this paper identifies the innovation potential of companies in financially undeveloped areas. While past research supports the spatial agglomeration and Matthew effect for company innovation, this paper provides a valuable reference to moderate the situation.

## Limitations and prospects

This study has some limitations. The results are limited to Chinese A-share listed companies that are comparably large and mature. It may be more difficult for small companies to obtain nonlocal loans, potentially meaning the influence of nonlocal loans is different depending on the company size. Moreover, there may be restrictions on the interregional circulation of credit resources in other countries, so the findings of this paper may not be widely applicable outside China.

Based on the limitations of this paper and the gaps in previous studies, further research could focus on the following aspects. First, nonlocal loans could be re-estimated with different boundary lines as the degree of market segmentation in the lending industry is likely to vary across countries. Second, while this paper establishes a link between nonlocal loan agreements and the external economic environment, further research could explore the determination of loan agreements using internal factors such as financial leverage, the government–company relationship, the company–bank relationship, and the hometown of executives. Third, the conditions of nonlocal loan agreements and the level of supervision from banking institutions could be discussed in greater depth. Finally, the economic effect of nonlocal loans could be explored. In particular, researchers could assess whether the loss of local credit resources obstructs economic development at the local level. The impact of nonlocal loans on the market share, financial performance, and investment preference of companies could also be tested.

## Supporting information

**S1 Dataset. Financial data of A-share listed companies in China from 2007 to 2022.**
(ZIP)

## Author Contributions

**Conceptualization:** Xinglin Liu.

**Data curation:** Yu Chen, Yan Xiong, Yanlin Wu.

**Formal analysis:** Yu Chen.

**Investigation:** Yanlin Wu.

**Methodology:** Yan Xiong.

**Supervision:** Xinglin Liu.

**Writing – original draft:** Xinglin Liu.

**Writing – review & editing:** Xinglin Liu.

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
