## [Decision Letter · Decision Letter 0]

8 Apr 2024

PONE-D-23-42502Interregional circulation of credit resources and company innovationPLOS ONE

Dear Dr. Liu,

Thank you for submitting your manuscript to PLOS ONE. After careful consideration, we feel that it has merit but does not fully meet PLOS ONE’s publication criteria as it currently stands. Therefore, we invite you to submit a revised version of the manuscript that addresses the points raised during the review process.

We look forward to receiving your revised manuscript.

Kind regards,

Hai Long

Academic Editor

PLOS ONE

2. PLOS requires an ORCID iD for the corresponding author in Editorial Manager on papers submitted after December 6th, 2016. Please ensure that you have an ORCID iD and that it is validated in Editorial Manager. To do this, go to ‘Update my Information’ (in the upper left-hand corner of the main menu), and click on the Fetch/Validate link next to the ORCID field. This will take you to the ORCID site and allow you to create a new iD or authenticate a pre-existing iD in Editorial Manager. Please see the following video for instructions on linking an ORCID iD to your Editorial Manager account: https://www.youtube.com/watch?v=_xcclfuvtxQ.

Additional Editor Comments:

The manuscript has to be English proofread by experts and attach the Editing certificate.

Reviewers' comments:

Reviewer's Responses to Questions

**Comments to the Author**

1. Is the manuscript technically sound, and do the data support the conclusions?

Reviewer #1: Partly

Reviewer #2: Yes

2. Has the statistical analysis been performed appropriately and rigorously? 

Reviewer #1: Yes

Reviewer #2: Yes

3. Have the authors made all data underlying the findings in their manuscript fully available?

Reviewer #1: Yes

Reviewer #2: Yes

4. Is the manuscript presented in an intelligible fashion and written in standard English?

Reviewer #1: Yes

Reviewer #2: No

5. Review Comments to the Author

Reviewer #1: The research addresses a relevant and significant topic. The title of the article accurately reflects its content and objectives. The introduction is comprehensive, providing context for the research and clearly stating its purpose. The study's goal is well-defined, and the individual sections are logically organized. The literature review is detailed and follows a logical order. The paper references 69 sources, primarily articles, which are robust and valuable, largely reflecting current research—a critical aspect for the paper's relevance. The cited works were not only appropriately selected but also critically analyzed. Citation of sources is accurate and consistent. The research methods employed by the author are suitable and empirically sound. The findings and conclusions presented offer significant insights for policymakers.

I recommend that this article be published after necessary minor improvements are made.

Here are some recommendations I would like to offer to the authors:

1. The abstract is very brief, reducing its informative value. Expanding the abstract to include the research methods would be beneficial.

2. References to the sources of statistical data must be provided.

3. Discussion should contain a more thorough comparison of the results of the conducted research with those of the preceding studies. It is important to indicate where the outcomes align with, or diverge from, previous research. They are currently written in a manner similar to a literature review.

4. The conclusions of the study need to be clarified. Specifically, the Conclusion section states: "The results prove that the nonlocal loans encourage companies to increase R&D investment.” However, it is crucial to note that a correlation study does not establish causality but rather identifies a statistical association.

5. It would be desirable to include DOI (if available) in the list of references for articles.

Reviewer #2: Please show the real problem , questions and objective in one section.

Add recent studies.

Present theoritical background

Present limitations

Present implication

Hypothesis should be rewritten , heading should only show variables

Present something about China industry

DV SECTION must be expanded.

Draw agraph showing relationships

Show roubtness test

6. PLOS authors have the option to publish the peer review history of their article (what does this mean?). If published, this will include your full peer review and any attached files.

Reviewer #1: No

Reviewer #2: No

---

## [Author Response · Author response to Decision Letter 0]

4 Jun 2024

Dear Editor(s) and Reviewer(s),

Thanks for your careful comments. We have attached great importance to them. Recently, the paper has been revised and submitted. The responses to your comments and the specific changes are presented as follows.

Responses to Reviewer 1

Comment:

The abstract is very brief, reducing its informative value. Expanding the abstract to include the research methods would be beneficial.

Response:

The content of the abstract has been extended. The Abstract emphasizes the difference between past research and this study at first. Then, the data source and methodology are briefly introduced. Next, research objectives and results are explained. Moreover, the results from the Discussion part highlight the role of nonlocal loans in supplementing disposable capital and the increase in innovation output. Finally, policy implications are provided. Details can be shown as follows.

Although the impact of interest rates, repayment periods, and loan scales on loan consequences has been extensively studied, little attention has been paid to the geographical distance involved in loan transactions. This study collects the addresses of borrowing companies, listed companies, and banks. Nonlocal loans can be distinguished because the regional segmentations in the lending industry reflect the features of provincial boundaries. Using data from Chinese A-share listed companies from 2007 to 2022, this research explores the causes of nonlocal loans and their impact on company innovation. Nonlocal loans address the lack of local credit resources rather than financial constraints, supplementing disposable capital. This interregional circulation of credit resources facilitates innovation, particularly in financially undeveloped areas. This study excludes the possibility of research and development (R&D) manipulation and recognizes the increase in innovation output. The findings have implications for credit resource allocation and balanced regional development.

Comment:

References to the sources of statistical data must be provided.

Response:

The data sources and sample composition are introduced in detail. Besides, the dataset related to this paper would be provided in the Supporting Information and submitted to the system.

Comment:

Discussion should contain a more thorough comparison of the results of the conducted research with those of the preceding studies. It is important to indicate where the outcomes align with, or diverge from, previous research. They are currently written in a manner similar to a literature review.

Response:

The comparison results between this study and other findings can be provided at the beginning of the Discussion part. Current research widely criticizes the drawbacks of nonlocal loans but does not explain their occurrence. This study acknowledges the disadvantages of nonlocal loans in terms of repayment period, interest rate, and scale. While other findings introduce the condition of credit traps, this study highlights the function of nonlocal loans as supplementary capital, especially in financially undeveloped areas. The interregional circulation of credit resources can promote more balanced and inclusive regional development. For the positive effect of nonlocal loans on company innovation, some challenges are posed while some literature demonstrates that companies in areas with sufficient credit obtain fewer nonlocal loans but more innovation output. This article does not emphasize that nonlocal loans have more advantages than local loans but rather confirms the financing ability of the companies and the supplement of nonlocal loans to disposable capital. Some literature provides similar viewpoints that the circulation of credit resources may facilitate the adoption and diffusion of new technologies. Please view the details in the revised manuscript.

Comment:

The conclusions of the study need to be clarified. Specifically, the Conclusion section states: "The results prove that the nonlocal loans encourage companies to increase R&D investment.” However, it is crucial to note that a correlation study does not establish causality but rather identifies a statistical association.

Response:

It may not be coherent to directly show the positive effect of nonlocal loans on company innovation. The structure of this paper can be adjusted. The cause of nonlocal loans can be placed in the baseline results. Then, the positive effect of nonlocal loans can be indicated. The findings suggest that companies are compelled to seek nonlocal loans when the provincial financial development level is comparably low. When the lack of local credit resources is eased, the supplementary funds of nonlocal loans encourage companies to increase R&D investment. Moreover, company innovation in financially developed areas does not highly rely on nonlocal loans, but the innovation enthusiasm of companies in financially undeveloped areas could be greatly mobilized by the injection of nonlocal loans. We improve the coherence and logic of the explanation in the Conclusion part.

Besides, we understand that the simultaneous presence of dependent and independent variables may imply a statistical association instead of a strong causality. Consequently, the value of a dependent variable in the next period can be estimated. Regressions in the revised manuscript obtain the effect of nonlocal loans on the current and future financing strategies of companies. This approach could be more understandable. The results are consistent with the original manuscript.

Furthermore, this paper excludes the possibility of R&D manipulation and emphasizes the growth in the innovation output, especially the invention patents. Overall, the causality between nonlocal loans and company innovation can be improved.

Comment:

It would be desirable to include DOI (if available) in the list of references for articles.

Response:

The DOI has been added to the references.

Responses to Reviewer 2

Comment:

Please show the real problem , questions and objective in one section.

Response:

The Introduction Section presents the background of market segmentation and the advocation of the cross-provincial allocation of credit resources at first. Then, it provides a brief literature review and concludes the research gaps. Next, the research objectives of this paper are introduced. The findings and contributions are reported consequently.

Comment:

Add recent studies.

Response:

The literature review is inadequate in the previous manuscript. More recent studies have been added to the Introduction part of the revised manuscript. Moreover, we supplement some findings about recent studies and compare the difference between them and our research in the Discussion part. Past literature widely criticizes the drawbacks of nonlocal loans but does not explain their occurrence. This study highlights the function of nonlocal loans as supplementary capital, especially in the financially undeveloped areas. Some challenges are posed while some literature demonstrates that companies in areas with sufficient credit obtain fewer nonlocal loans but more innovation output. It is noteworthy that this article does not emphasize that nonlocal loans have more advantages than local loans but rather confirms the financing ability of the companies and the supplement of nonlocal loans to disposable capital.

Comment:

Present theoritical background

Response:

We have adjusted the Hypothesis Section, which mainly includes the causes of nonlocal loans, the effect of nonlocal loans on company innovation, and the influence of interest rate liberalization policy. The analysis is based on the resource dependence theory. The theory posits that organizations are interdependent with their external environment. They can adjust the degree of this dependence through various strategic options. In the context of corporate financing, companies may seek nonlocal bank loans if the credit resources provided by local banking institutions are insufficient to meet their operational or innovative needs. Then, we analyze the economic consequences of nonlocal loans.

When explaining the results, we refer to the polarization theory. Without external intervention, the polarization effect will dominate the market and trigger the stratification of the economy. However, the findings of this paper reject the application of the polarization policy in China and confirm the interregional circulation of credit resources. The positive effect of nonlocal loans on companies in financially undeveloped areas helps to relieve the imbalance in regional economic development.

The economic background has been reported in the Introduction Section. We highlight the market segmentation in China. Referring to the “Guideline on Accelerating the Construction of Shanghai International Financial Center and Providing Financial Support for the Integrated Development of the Yangtze River Delta”, the Chinese government tends to encourage the cross-provincial allocation of credit resources to balance the regional development. Hence, we introduce the topic of this paper.

Comment:

Present limitations

Response:

The limitations of past literature can be found in the Introduction part. Firstly, past research explores the effect of loans on innovation in terms of interest rate, loan scale, repayment period, and credit market accessibility [16, 17]. The geographic distance for loan business has received limited consideration. Secondly, the shortage of remote loan business has been emphasized while the reasons that companies obtain the loans have not been discussed deeply.

The limitations of this paper are presented in the Limitations and prospects Section. The results are limited to Chinese A-share listed companies which are comparably large and mature. It could be more difficult for small companies to obtain nonlocal loans, then the effect of nonlocal loans might be questionable. Moreover, there might be some restrictions on the interregional circulation of credit resources in other countries, so the situation in this paper may not be widely applicable.

Comment:

Present implication

Response:

The implication of this paper is reported in the Conclusions and implications Section. The long geographic distance increases the difficulty in evaluating the loan default risk of the companies. Closer cooperation is required to be built for the banks in different regions. Sharing the regulatory information could be efficient and rigorous to identify high-quality companies in financially undeveloped areas, which helps banks provide financial support pertinently. In addition, the collaboration between banking institutions and China National Intellectual Property Administration could be improved, which makes it easier for banks to recognize innovative companies with great potential in financially undeveloped areas.

Comment:

Hypothesis should be rewritten , heading should only show variables

Response:

Both the contents and the headings of the Hypothesis development Section have been modified. We decide to propose the cause of nonlocal loans at first, from the perspective of financial development level and financing constraints. Then, the effect of nonlocal loans on company innovation can be analyzed. Next, the influence of loan interest rate liberalization policy can be considered. The logic chain may be more fluent. 

The analysis might be comparably ambiguous for the positive and negative effects of nonlocal loans on company innovation in the original manuscript. We have carefully adjusted the expression and made differentiated analyses.

The overall expression of the Hypothesis development Section can be more coherent in the revised manuscript. 

Comment:

Present something about China industry

Response:

Admittedly, industry research can reflect the condition of loan businesses in detail. We have tried to make some pie graphs to show the statistical results by industry and year. The results are strongly consistent in that the manufacturing and real estate industries have a remarkable proportion of loan businesses. Besides, the heterogenous analysis based on industries cannot be appropriately explained as the significance of the core explaining variable varies. Overall, we decide to describe the comprehensive condition of the loan business in China. In Figure 1, the shrink of loan business from 2019 to 2021 can be explained by the economic downturn caused by the Corona Virus Disease 2019. With the economic recovery in China, the expansion of the loan business has happened in 2022.

In the Introduction Section, we highlight the special policy announced by the Chinese government. In February 2020, China issued a “Guideline on Accelerating the Construction of Shanghai International Financial Center and Providing Financial Support for the Integrated Development of the Yangtze River Delta” (Guideline). The Guideline encourages the cross-provincial allocation of credit resources. It aims to balance and meet the credit needs of companies in various provinces. Based on that, this paper sheds light on the interregional circulation of credit resources. 

Comment:

DV SECTION must be expanded.

Response:

We place the cause of nonlocal loans in the baseline results. Then the content of the Discussion could be insufficient. We estimate the effective paths of nonlocal loans and observe the results of R&D manipulation and R&D output. This Section further strengthens the findings in the baseline results. The baseline results indicate that the main cause of nonlocal loans is the lack of local credit resources. The Discussion Section confirms that the capital replenishments stimulate companies in financially undeveloped areas to add R&D investment, which helps to relieve the imbalance in regional economic development. Moreover, the increase in R&D investment is not an opportunistic behavior for tax incentives. The injection of nonlocal loans contributes to more innovation output, especially for invention patents. This article does not emphasize that nonlocal loans have more advantages than local loans but rather highlights the financing ability of the companies and the supplement of nonlocal loans to disposable capital.

Comment:

Draw a graph showing relationships.

Response:

The information might be insufficient if a graph is designed to show the significance of coefficients only. Then, we decide to draw a graph to indicate the borrowing situation of Chinese companies from 2007 to 2022. It can be found in the Descriptive results. Figure 1 intuitively reports the growth and the fluctuation of loan businesses in China.

For the overall trend, credit resources have been increasingly important to company financing. The number of companies with loan business experienced considerable growth in 2013. The loan interest rate liberalization policy has led to fewer financing costs since 2013. Then, companies are more willing to participate in the loan business. The shrink of loan business from 2019 to 2021 can be explained by the economic downturn caused by the Corona Virus Disease 2019. With the economic recovery in China, the expansion of the loan business has happened in 2022. The comparison results in Figure 1 indicate that companies are less likely to receive nonlocal loans. Although the policy has deepened the financial reform, the market segmentation in the loan business still exists.

Comment:

Show roubtness test.

Response:

The original manuscript sets the robustness test after the baseline results in the Main empirical results. It may be a misunderstanding. Then, we place the part in a separate section in the revised manuscript. The robustness tests contain the Reciprocal causation test, Selection bias test, and Alternative estimation of nonlocal loans. Results support the main findings of this paper.

Apart from the modification above, some adjustments have been made. Firstly, the revised manuscript has been adjusted to meet PLOS ONE's style requirements. Secondly, the ORCID ID for the corresponding author has been supplemented. Thirdly, the dataset has been uploaded. Fourthly, we have polished the paper and improved the overall coherence of the writing. The manuscript has been English proofread by experts. The Editing certific

---

## [Decision Letter · Decision Letter 1]

16 Jul 2024

PONE-D-23-42502R1Interregional circulation of credit resources and company innovationPLOS ONE

Dear Dr. Liu,

Thank you for submitting your manuscript to PLOS ONE. After careful consideration, we feel that it has merit but does not fully meet PLOS ONE’s publication criteria as it currently stands. Therefore, we invite you to submit a revised version of the manuscript that addresses the points raised during the review process.

**ACADEMIC EDITOR: **Two reviewers have accepted the manuscript for publication, but one of the reviewers suggested an English proofreading. Please make an English editing with a professional institution and resubmit it with an English editing certificate.

We look forward to receiving your revised manuscript.

Kind regards,

Hai Long

Academic Editor

PLOS ONE

Journal Requirements:

Reviewers' comments:

Reviewer's Responses to Questions

**Comments to the Author**

1. If the authors have adequately addressed your comments raised in a previous round of review and you feel that this manuscript is now acceptable for publication, you may indicate that here to bypass the “Comments to the Author” section, enter your conflict of interest statement in the “Confidential to Editor” section, and submit your "Accept" recommendation.

Reviewer #1: All comments have been addressed

Reviewer #2: All comments have been addressed

2. Is the manuscript technically sound, and do the data support the conclusions?

Reviewer #1: Yes

Reviewer #2: Yes

3. Has the statistical analysis been performed appropriately and rigorously? 

Reviewer #1: Yes

Reviewer #2: Yes

4. Have the authors made all data underlying the findings in their manuscript fully available?

Reviewer #1: Yes

Reviewer #2: (No Response)

5. Is the manuscript presented in an intelligible fashion and written in standard English?

Reviewer #1: Yes

Reviewer #2: Yes

6. Review Comments to the Author

Reviewer #1: The author has taken my comments into account. The manuscript meets the journal's requirements. I recommend the article for publication.

Reviewer #2: Required changes are met. The checking of English should be verfied. The writing style must be also checked by the journal

7. PLOS authors have the option to publish the peer review history of their article (what does this mean?). If published, this will include your full peer review and any attached files.

Reviewer #1: No

Reviewer #2: No

---

## [Author Response · Author response to Decision Letter 1]

25 Jul 2024

Dear Reviewer(s),

Thanks for your careful comments. We have attached great importance to them. Recently, the paper has been revised and submitted. The responses to your comments and the specific changes are presented as follows.

Responses to Reviewer 1

Comment:

The author has taken my comments into account. The manuscript meets the journal's requirements. I recommend the article for publication.

Response:

Thanks for your positive feedback on the manuscript. We appreciate you taking the time to provide constructive comments during the review process. It has helped us to improve the quality and clarity of the work.

Responses to Reviewer 2

Comment:

Required changes are met. The checking of English should be verfied. The writing style must be also checked by the journal.

Response:

We have adopted proofreading and editing services. Two editors from Cambridge Proofreading LLC have corrected the grammar and improved the readability of the text. The proofreading and editing certificates have been submitted to the system as separate files labeled 'Other'.

Moreover, we have made some adjustments to ensure the manuscript meets PLOS ONE's style requirements. We have checked the heading type and font. We have used the PACE tool to check the figure and convert it to the accepted format. Then, we have uploaded the figure separately. Consistent with the requirements, the manuscript can be presented in double-spaced paragraph format. Additionally, we have changed the in-text citations from the hyphen (-) to the en-dash (–), e.g., [1-2] -> [1–2]. As for references, the journal name should be used as an abbreviation. We have modified it. The Supporting information is required to be listed at the end of the manuscript in the template file. But it appears before the References in the published articles. After careful consideration, we ensure that the revised manuscript meets the requirements of the PLOS ONE style templates. The templates can be found at the links below.

We refer to published literature for some formats not mentioned in the template files. For example, The first letter is capitalized for “Supporting information”, while all words have their first letters capitalized for “Author Contributions”.

Thanks for your valuable feedback. It enhances the clarity and comprehensiveness of our research. It also assists us in fulfilling the necessary requirements of the journal more effectively. We sincerely hope that the revised manuscript can be accepted for publication.

Kind regards,

Xinglin Liu

---

## [Editor Report · Decision Letter 2]

1 Aug 2024

Interregional circulation of credit resources and company innovation

PONE-D-23-42502R2

Dear Dr. Liu Xinglin,

We’re pleased to inform you that your manuscript has been judged scientifically suitable for publication and will be formally accepted for publication once it meets all outstanding technical requirements.

Kind regards,

Hai Long

Academic Editor

PLOS ONE
---

## [Editor Report · Acceptance letter]

9 Aug 2024

PONE-D-23-42502R2 

PLOS ONE

Dear Dr. Liu, 

I'm pleased to inform you that your manuscript has been deemed suitable for publication in PLOS ONE. Congratulations! Your manuscript is now being handed over to our production team.

Kind regards, 

on behalf of

Dr. Hai Long 

Academic Editor

PLOS ONE